# Hierarchical Co(OH)_2_ Dendrite Enriched with Oxygen Vacancies for Promoted Electrocatalytic Oxygen Evolution Reaction

**DOI:** 10.3390/polym14081510

**Published:** 2022-04-08

**Authors:** Tingting Zhou, Zhen Cao, Xishi Tai, Lei Yu, Jian Ouyang, Yunfei Li, Jitao Lu

**Affiliations:** 1College of Chemical Engineering and Environmental Chemistry, Weifang University, Weifang 261061, China; zhoutingting1986@163.com (T.Z.); jimiyulei@163.com (L.Y.); 2Shenzhen Kohodo Hydrogen Energy Co., Ltd., Shenzhen 518109, China; ouyangjian@kohodo.cn (J.O.); liyunfei@kohodo.cn (Y.L.)

**Keywords:** electrocatalyst, dealloyed, oxygen vacancy, hierarchical structure, oxygen evolution reaction

## Abstract

It is critical to develop efficient oxygen evolution reaction (OER) catalysts with high catalytic properties for overall water splitting. Electrocatalysts with enriched vacancies are crucial for enhancing the catalytic activity of OER through defect engineering. We demonstrated the dealloying method in a reducing alkaline solution using the Co_5_Al_95_ alloy foil as a precursor to produce a new oxygen-vacancy-rich cobalt hydroxide (O_V_−Co(OH)_2_) hierarchical dendrite. The as-synthesised O_V_−Co(OH)_2_ showed superior electrocatalytic activities toward OER when compared to pristine cobalt hydroxide (*p*–Co(OH)_2_), which had a low onset overpotential of only 242 mV and a small Tafel slope of 64.9 mV dec^−1^. Additionally, for the high surface area provided by the hierarchical dendrite, both *p*–Co(OH)_2_ and O_V_−Co(OH)_2_ showed a superior activity as compared to commercial catalysts. Furthermore, they retained good catalytic properties without remarkably decaying at an overpotential of 350 mV for 12 h. The as-made O_V_−Co(OH)_2_ has prospective applications as an anode electrocatalyst in electrochemical water-splitting technologies with the advantages of superior OER performances, large surface area and ease of preparation.

## 1. Introduction

Hydrogen (H_2_), a prospective clean energy source, can be easily generated through electrochemical water splitting with zero greenhouse gas emissions, hence enhancing clean energy sustainability [1]. However, because electrocatalytic processes involve oxygen atoms, the oxidative half-reaction–oxygen evolution reaction (OER) is considered a bottleneck process in electrochemical water splitting due to its slow kinetics and high overpotential, which results in a significant decrease in energy utilisation efficiency [2,3,4]. The key to resolving this issue is to explore efficient catalysts for reaction acceleration.

While noble metal catalysts, such as Ir/Ru or their oxides, are now viable electrocatalysts for OER, their scarcity, high cost and poor operating stability obstruct their scale-up application [5,6]. As a result, great strides have been made in utilising earth-abundant transition metal elements as prospective OER electrocatalysts because of their promising activity and durability [7]. Among various potential materials, Co-based materials possess the appropriable valence states, and structural flexibility has been proposed as a new generation of highly effective OER electrocatalysts [7,8]. Generally, when developing high-efficiency catalysts, the number of exposed active sites and intrinsic activities should be considered [9]. However, most Co-based materials fell short of meeting actual application requirements because of their inadequate intrinsic activities and limited surface area. Additionally, as the intrinsic activity of cobalt hydroxides is largely determined by the electronic structures of the Co atoms, a variety of strategies have been used to modify the surface oxidation states, including introducing defect sites [10,11] and hybridisation [12,13]. An oxygen vacancy (Ov) is a type of inherent defect found in Co-based materials. Xiao et al. investigated the effect of an oxygen vacancy on Co_3_O_4_ with a high oxygen-vacancy concentration during the OER process by various operando characterizations, demonstrating that the oxygen vacancy facilitated the formation of surface CoOOH, which is valuable for boosting the OER activity [14]. Generally, the CoOOH is considered the active site for OER [15,16]. Theoretical calculations show that decreasing binding energy for the formation of intermediates can significantly increase the reactivity of the active site [17]. Apart from the intrinsic activity of active sites, the number of active sites connected with the structures and morphology of nanomaterials is also important for the OER process. Previous studies have established that hierarchical architecture can provide several benefits, such as a larger surface area, increased permeability and higher porosity, resulting in a considerable improvement in electrochemical catalysis [18,19,20].

According to the above considerations, it is necessary to design a strategy for simultaneously creating oxygen vacancies and a hierarchical structure in cobalt hydroxide to intrinsically improve the intrinsic OER activity and the number of active sites. In this study, we described the facile reduction–dealloying synthesis of oxygen-vacancy-rich cobalt hydroxide (O_V_−Co(OH)_2_) with a hierarchical dendrite structure. Although both of the O_V_−Co(OH)_2_ and pristine cobalt hydroxide (*p*–Co(OH)_2_) have similar nano-structures, the O_V_−Co(OH)_2_ NCAs possessed a lower onset potential and Tafel slope, showing that O_V_ had a positive effect on intrinsic activity. Significantly, both O_V_−Co(OH)_2_ and *p*–Co(OH)_2_ showed a superior activity in comparison to commercial electrocatalysts, which can be attributed to the unique hierarchical dendrite structure, which provides a larger surface area and is convenient for the penetration of electrolytes.

## 2. Experimental Section

### 2.1. Chemical Reagents

Sodium borohydride (NaBH_4_), sodium hydroxide (NaOH), potassium hydroxide (KOH) and ethanol (CH_3_CH_2_OH) were supplied by Sinopharm Chemical Reagent Co. Ltd, Shanghai, China. Nafion (5 wt.%) was supplied by Shanghai Branch, Du Pont China Holding Co., Ltd., Shanghai, China. The XC-72 was supplied by Shanghai Hesen Electric Co. Ltd., Shanghai, China. The commercial IrO*_x_* and Pt/C were supplied by Alfa Aesar, Ward Hill, MA, United States.

### 2.2. Synthesis of O_V_−Co(OH)_2_ and p−Co(OH)_2_

The Co_5_Al_95_ alloy foil was prepared using the melt-spinning technique in a controlled argon-protected atmosphere. The reduction–dealloying procedure was performed in 100 mL of 2 M NaOH solution containing NaBH_4_ at 30 °C until no visible bubbles emerged (about 12 h), then soaked in 100 mL of 0.1 M NaOH solution to remove residual Al; O_V_−Co(OH)_2_ was then obtained. Finally, thorough washing with distilled water many times and drying under a vacuum were carried out. The *p*–Co(OH)_2_ was prepared in 2 M NaOH solution without adding NaBH_4_ with other factors fixed.

### 2.3. Materials Characterisation

The crystal phases were characterised using a SmartLab diffractometer equipped with CuK*α* radiation. The morphology was studied using scanning electron microscopy (SEM), ZEISS MERLIN. The microstructure investigations were performed using transmission electron microscopy (TEM, FEI Tecnai G2 T20, Stanford Nano Shared Facilities, Stanford, CA, USA). The electron spin resonance (ESR) spectra were obtained via Bruker EMXplus. The X-ray photoelectron spectroscopy (XPS) measurements were performed on a PHI 5000 VersaProbe. The porous structural parameters were determined using Brunauer–Emmett–Teller (BET, ASAP 2460).

### 2.4. Electrochemical Measurements

Electrochemical experiments were performed in a three-electrode cell using a CHI 760E electrochemical workstation (CH Instruments, Inc., Shanghai, China) at 25 °C in an O_2_-saturated 1 M KOH solution. A Hg/HgO electrode and graphite rod were used as the reference and counter electrodes, respectively. The working electrode was prepared as follows: 4.0 mg electrocatalyst powder and 3.0 mg XC-72 were dispersed in 420 µL ethanol, 500 µL water and 80 μL Nafion solution (5 wt.%), then sonicated for 30 min. On the glassy carbon surface of a revolving disc electrode (RDE), 10 μL catalyst suspension was placed. Except for XC-72, all commercial catalyst electrode samples were prepared using the same method. All scanning cyclic voltammogram (CV) and linear sweep voltammetry (LSV) measurements except for CVs of electrochemical surface area (ECSA) were performed at a sweep rate of 5 mV s^−1^ with a 75% iR-compensation. Durability was assessed using the controlled potential electrolysis method. Electrochemical impedance spectroscopy (EIS) was performed at 1.55 V with an amplitude of 5 mV over a frequency range of 10^−1^–10^5^ Hz. The values of the double-layer capacitance (Cdl) were determined by scanning CV curves at various scan rates in the range of 0–0.05 V vs. Hg/HgO. All potential values were calibrated in terms of a reversible hydrogen electrode vs. a non-reversible hydrogen electrode (RHE).

## 3. Results and Discussion

Oxygen vacancies enriched O_V_−Co(OH)_2_ with hierarchical dendrite structure were synthesised through a reduction-dealloying method, as shown in Figure 1. The *p*–Co(OH)_2_ was performed in the same concentration of NaOH solution without NaBH_4_. The addition of NaBH_4_ significantly regulated the dealloying process and had an impact on the crystal structures and morphologies of the final sample during this process. The O_V_−Co(OH)_2_ demonstrated a hierarchical structure of the dendrite nano-structure tightly packed with tiny flakes, as shown in the SEM images (Figure 1) whereas the *p*–Co(OH)_2_ showed a similar hierarchical structure, but the number of nanoflakes was significantly increased. The large number of flakes generated during the pristine dealloying process was partially agglomerated into nanoflake clusters due to the high activity of Co endowed by the pristine alkaline solution with a higher oxidative capacity. Unlike pristine alkalines that are highly oxidative, reduction alkalines containing NaBH_4_ could provide an in situ reduction micro-environment throughout the corrosion process. Although NaBH_4_ could not entirely impede the oxidation of Co in the Co_5_Al_95_ alloy foil, it acted as an auxiliary and resulted in just a few flakes tightly packed on a dendrite nano-structure in O_V_−Co(OH)_2_. It is worth noting that the latter morphology not only improved the external surface area but also prevented excess flakes from blocking the pores, impeding electron transportation.

The structural features of the Co_5_Al_95_ alloy foil and dealloying products were detected using X-ray diffraction (XRD). The XRD patterns of the Co_5_Al_95_ alloy foil corresponded to those of Al_9_Co_2_ (JCPDS No. 65-6460) and Al (JCPDS No. 65-2869), as shown in Appendix A. As shown in Figure 2a, after dealloying, the XRD patterns of *p*–Co(OH)_2_ could be indexed to Co(OH)_2_ (JCPDS No. 45-0031) and a trace of Co_6_CO_3_(OH)_16_·4H_2_O (JCPDS No. 51-0045), showing that Co formed into Co(OH)_2_. By comparing the XRD patterns of Ov−Co(OH)_2_ and *p*–Co(OH)_2_, it could be seen that the degree of crystallisation of the Ov−Co(OH)_2_ clearly decreased relative to the *p*–Co(OH)_2_ as shown by the weakened intensity of the diffraction peaks in the Figure 2a, which might be due to the introduction of NaBH_4_. The peaks corresponding to Co(OH)_2_ (JCPDS No. 45-0031) were almost invisible and only a few peaks corresponding to Co_6_CO_3_(OH)_16_·4H_2_O (JCPDS No. 51-0045) were partially retained. The tiny amounts of Co_6_CO_3_(OH)_16_·4H_2_O in these samples were probably derived from atmospheric carbon dioxide (Appendix A). The low-magnification TEM images in Figure 2b,c confirm that a branch of dendrite had an average length of 3–4 μm and a width of 1–2 μm. More importantly, the high-resolution transmission electron microscopy images (HRTEM) in Figure 2d,e clearly show the presence of a nanoporous structure with pore sizes ranging from 3 to 5 nm in the dendrite section, as well as covered nanoflakes with a thickness of ~3 nm. The unique dendrite hierarchical structure is beneficial for the exposure of active sites and transportation of electrolytes, resulting in a superior electrochemical performance. The lattice fringes of nanoporous dendrite structure are clearly visible, showing that the interplanar distances of 0.24 and 0.23 nm correspond to the (101) and (002) plane of the Co(OH)_2_ phase(JCPDS No. 45-0031), respectively (inset 1, 2 of Figure 2d). The lattice spacing of nanoflakes shows an interplanar distance of 0.28 nm, which is consistent with the (110) plane of typical Co(OH)_2_ (JCPDS No. 45-0031, inset 1 of Figure 2e).

The surface area and mesoporous characteristics of the material were determined using BET surface area measurements (Figure 2f). The O_V_−Co(OH)_2_ show a larger BET specific surface area (215.6 m^2^/g) than *p*–Co(OH)_2_ (176.1 m^2^/g) and a similar pore size of ~3 nm, which is consistent with the TEM results. This comparison confirms that introducing NaBH_4_ during the dealloying process increases the surface area of Co(OH)_2_, resulting in an increase in active sites and abundant channels for the electrocatalytic process [10]. Additionally, a distinct hysteresis loop with a typical type-IV adsorption–desorption isotherm was observed, showing the presence of a nanosheet structure [21]. To gain an insight into the ECSA, the CV curves at various scan rates were used to determine the electrochemical double-layer capacitance (Cdl) (Appendix A). It was worth noting that the Cdl value of O_V_−Co(OH)_2_ was 3.38 mF cm^−2^, which was greater than that of *p*–Co(OH)_2_ (2.05 mF cm^−2^). The superior BET surface area and ECSA values show that the unique dendrite nano-structure packed with small flakes, as well as the resulting abundant active sites, are likely another important factor for promoting OER electrochemical activity.

XPS was used to investigate the electronic states of Co and O on the surface of various materials (Figure 3a,b). For *p*–Co(OH)_2_, the Co 2p XPS spectra showed that the peaks of 2p1/2 and 2p3/2 were located at 796.8 and 780.7 eV, respectively. A peak fitting analysis showed that the Co oxidation state was primarily Co^2+^ with a small quantity of Co^3+^ [22]. In addition, the XPS spectra of Ov−Co(OH)_2_ were similar to those of *p*–Co(OH)_2_. However, the binding energy of Co 2p was ~0.6 eV higher in O_V_−Co(OH)_2_ than in *p*–Co(OH)_2_. Specifically, the splitting peaks of 781.4 and 780.6 eV were derived from Co 2p_3/2_ and attributed to Co^3+^ and Co^2+^, and the peaks of 797.5 and 796.0 eV were derived from Co 2p_1/2_ [20,23,24]. In order to clarify the state of Co, we also analysed the XPS spectra of C 1s (Appendix A). The spectra of C 1s could also be split into peaks at binding energies of 288.8, 285.6 and 284.7 eV corresponding to satellite peak, C–OH and C–O bonds [25]. Herein, the chemical states were further understood by XRD and XPS; Ov−Co(OH)_2_ and *p*–Co(OH)_2_ was a mixture of hydroxide and carbonate. It should be mentioned that the two samples exhibited comprehensive performances of the mixture they contained when performing OER. As shown in the O 1s spectra of these two samples (Figure 3b), three typical peaks at 533.0, 531.3 and 530.5 eV could be fitted, corresponding to H_2_O adsorbed on the surface, hydroxide oxygen/carbonate and oxygen vacancies [24,26,27]. Additionally, the O_V_−Co(OH)_2_ had a substantially higher surface oxygen vacancies ratio than *p*–Co(OH)_2_. Moreover, enhanced O−vacancy could result in hybridisation between the Co 3d and O 2p orbitals [28]. Electron spin resonance (ESR) spectra were employed to confirm their electronic structures (Figure 3c). Both produced a symmetrical signal at g = 2.003, showing that the electrons were trapped in oxygen vacancies [29]. We can conclude from the XPS and ESR results that the introduction of NaBH_4_ during the dealloying process results in a sharp increase in oxygen vacancies and Co^2+^, which may significantly improve the electronic structures, and thus OER activity.

The electrocatalytic behaviours were investigated using CV curves, as shown in Figure 4a. Two redox couples were assigned to the Co^3+^/Co^2+^ and Co^4+^/Co^3+^ redox in the CV curves of *p*–Co(OH)_2_ and O_V_−Co(OH)_2_, respectively, at 1.08 and 1.42 V vs. RHE [30]. The broadened double-layer and negatively-shifted redox couplings showed that the O_V_−Co(OH)_2_ possessed a larger capacity area, which was not only due to the increase of specific surface, but also likely related to the existence of oxygen vacancy [31]. The presence of vacancies had a specific oxidation effect on Co−based materials, which favoured OER [14]. The electrocatalytic activity towards OER was determined by the LSV curves, as shown in Figure 4b. The O_V_−Co(OH)_2_ had a lower onset overpotential of 242 mV for OER than the O_V_−Co(OH)_2_ (250 mV), as well as a significantly faster increase in current density. As a result, the O_V_−Co(OH)_2_ required a lower overpotential of 350 mV than the *p*–Co(OH)_2_ (375 mV) to achieve a current density of 10 mA cm^−2^. These results showed that the introduction of O_V_ significantly improved the electrocatalytic performance of OER. Commercial Pt/C did not show an OER catalytic activity, but commercial IrO*_x_* showed a benchmarking level of OER activity. Kinetic analyses of OER were performed using Tafel plots and EIS. The Tafel slope of the O_V_−Co(OH)_2_ (64.9 mV dec^−1^) was smaller than those of *p*–Co(OH)_2_ (66.1 mV dec^−1^) and IrO*_x_* (81.1 mV dec^−1^), as shown in Figure 4c, the introduction of O_V_ enhances the reaction rate and electrocatalytic kinetics. Surprisingly, both O_V_−Co(OH)_2_ and *p*–Co(OH)_2_ showed superior electrocatalytic activities and kinetics when compared to commercial IrO*_x_*, which should be attributed to the similar dendrite hierarchical structure providing an abundance of active sites. Simultaneously, as shown in the Nyquist plots of EIS (Figure 4e), a lower charge-transfer resistance is observed for O_V_−Co(OH)_2_, showing a more efficient charge transport of the O_V_−Co(OH)_2_ during the electrocatalytic process. These results showed that the kinetics of OER were enhanced by the introduction of oxygen vacancies and the formation of a dendrite hierarchical structure. The stability of O_V_−Co(OH)_2_ was further determined using chronopotentiometry, as shown in Figure 4f. After 12 h of testing at 350 mV overpotential, the current attenuation was less than 4%, supporting the stability of the O_V_−Co(OH)_2_. Altogether, these results showed that the O_V_−Co(OH)_2_-containing oxygen vacancies had an effective and stable OER performance.

## 4. Conclusions

A simple but efficient reduction-dealloying method was used to successfully synthesise O_V_−Co(OH)_2_ with a hierarchical dendrite structure. The superior catalytic efficiency of O_V_−Co(OH)_2_ for OER generated a low-onset overpotential of only 242 mV and a small Tafel slope of 64.9 mV dec^−1^. The excellent activity could be attributed to two factors: first, the oxygen vacancy in the O_V_−Co(OH)_2_ increased the intrinsic activity of the Co−active sites; second, the dendrite’s hierarchical nano-structure resulted in an abundance of specific surface area that was convenient for electrolyte penetration. Thus, the reduction-dealloying method enabled the simultaneous realisation of these two merits, making it a potential approach for the fabrication of other hierarchical nanostructures for the next generation of oxygen-related materials.

## Figures and Tables

**Figure 1 polymers-14-01510-f001:**
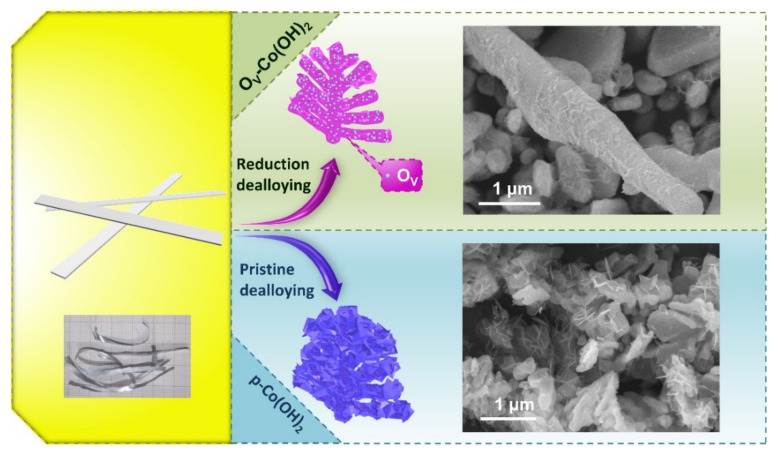
Schematic illustration and scanning electron microscopy images of the synthetic strategy of O_V_−Co(OH)_2_ and *p*–Co(OH)_2_.

**Figure 2 polymers-14-01510-f002:**
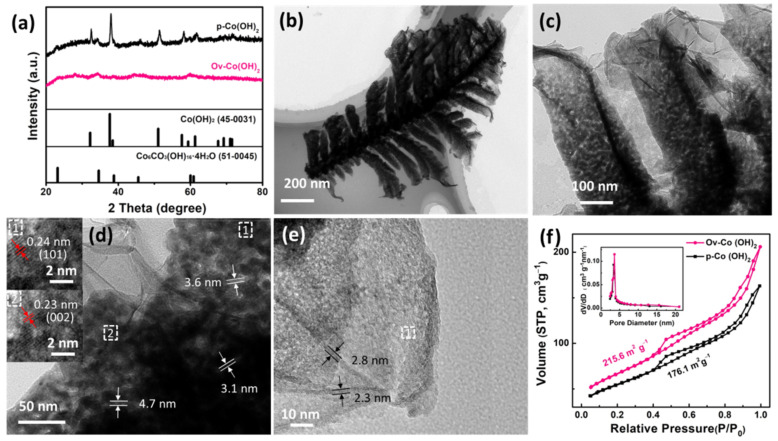
(**a**) X-ray diffraction patterns of *p*–Co(OH)_2_ and O_V_−Co(OH)_2_; (**b**,**c**) transmission electron microscopy images of O_V_−Co(OH)_2_; (**d**) high-resolution transmission electron microscopy (HRTEM) images of the dendrite section of O_V_−Co(OH)_2_; (**e**) HRTEM images of the covered nanoflakes of O_V_−Co(OH)_2_; (**f**) N_2_ adsorption and desorption isotherms and the corresponding pore size distribution (inset) of O_V_−Co(OH)_2_ and *p*−Co(OH)_2_.

**Figure 3 polymers-14-01510-f003:**
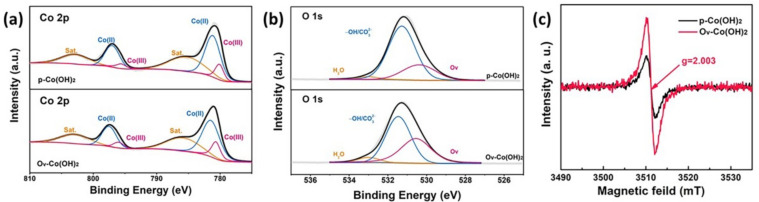
X-ray photoelectron spectra of Co 2p (**a**) and O1s (**b**) for *p*–Co(OH)_2_ and O_V_–Co(OH)_2_; (**c**) electron spin resonance spectra of O_V_–Co(OH)_2_ and *p*–Co(OH)_2_.

**Figure 4 polymers-14-01510-f004:**
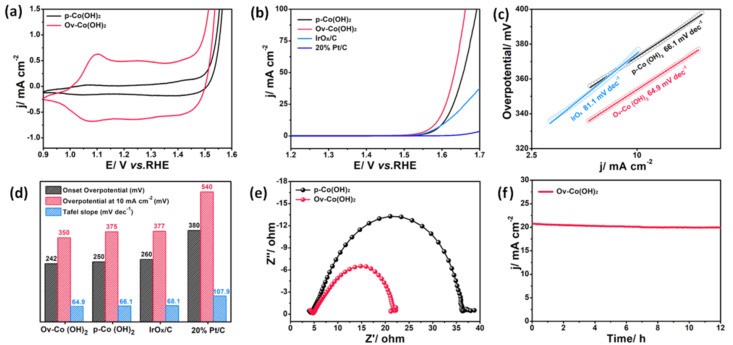
(**a**) Cyclic voltammetry curves of O_V_−Co(OH)_2_ and *p*–Co(OH)_2_; (**b**) linear sweep voltammetry curves of O_V_−Co(OH)_2_, *p*–Co(OH)_2_, IrO*_x_* and Pt/C; (**c**) corresponding Tafel slopes of O_V_−Co(OH)_2_, *p*–Co(OH)_2_ and IrO*_x_*; (**d**) comparison of oxygen evolution reaction catalytic parameters O_V_−Co(OH)_2_, *p*–Co(OH)_2_, IrO*_x_* and Pt/C; (**e**) Nyquist plots of O_V_−Co(OH)_2_ and *p*–Co(OH)_2_; (**f**) chronopotentiometric curve at the overpotential of 350 mV for O_V_−Co(OH)_2_.

## Data Availability

Not applicable.

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
