# Peer review of "Hierarchical Co(OH)2 Dendrite Enriched with Oxygen Vacancies for Promoted Electrocatalytic Oxygen Evolution Reaction"

_polymers, 2022, doi:10.3390/polym14081510_

Round 1
Reviewer 1 Report
The paper deals with the development of a so-called oxygen-vacant Cobalt hydroxide as electrode materials for OER in alkaline medium. Despite of the undeniable importance of the subject the paper shows a significant amount of problems that impels me to reject it for publication.
1. The dealloyng procedure applied is able to retain Al on the surface. The same procedure was already applied to obtain PtSn surface material a few decades ago. I strongly suggest performing an elemental analysis of the surface. By the way, the X-ray diffractogram displayed in supplementary material reinforces such issue.
2. In lines 146-147 the authors state that the peaks for p-Co(OH)2 and Ov-Co(OH)2 are identical. This is obviously wrong.
3. CV’s in S3 shows clearly that the material presents a very pronounced non-compensated iR.
4. Were the EIS plots corrected for ESCA?
5. In lines 213-214 the authors concluded from CV’s that the material has more oxigen-deficient sites. How to explain such assumption?
6. How could you explain the presence of broad peak in the CV as due to the oxygen vacancy? Could it not be due the non-dealloyed Al?
7. The decreasing on the onset potential by 8mV is not significant to attest any actual performance improvement.
8. The overpotential decreasing(25mV) is also non-significant to attest an important performance improvement.
Reviewer 2 Report
Zhou and coworkers demonstrated a hierarchical Co(OH)2 dendrite by pulse electrodeposited method. This paper demonstrated that the abundance of oxygen vacancies may lead to a large number of surface active sites, contributing to enhanced catalytic activity. While this work is interesting, there are ambiguities and deficiencies in the article that need to be improved and corrected. Before considering the suitability of a paper for publication in Polymer, the authors should consider the following points:
1. In the section of Experimental, it is suggested that the authors supplement details of the synthesis time and temperature, as well as the volume of reagents.
2. In the section of Introduction, there are errors and poor wording in the author's statements. In Paragraph 1, the authors claimed that “However, because electrocatalytic processes involve oxygen…….” is conceptually unclear, with O2 being the anodic product rather than the reactant in the catalytic process of Water Splitting. In Paragraph 2, robust durability (poor wording) and exceptional (exaggeration) is suggested that the authors correct them.
3. In Figure 3, the peaks splitting in XPS spectrum are wrong. It is suggested that the authors correct them.
4. The analysis of the physical phase of the catalyst is unclear. In Figure 2a, the XRD peaks do not correspond well to the JCPDS standard cards and it is suggested that the authors to correct them.
5. Which characterisation was used to demonstrate 25-μm thickness? It is suggested that the authors to supplement AFM tests.
6. The OER overpotential of commercial IrO2 is much higher than that reported in the literatures. It is suggested that the authors calibrate the reference electrode and retest it.
Reviewer 3 Report
The work explores Co-based electrocatalysis for OER in the alkaline solution. I rated the novelty of the work as average because the role of oxygen vacancies to the ORR/OER has been widely discussed and explored by hundreds of publications. The performance reported in this work is not in the top tier of this field. The good thing is that the work uses nanorod morphology for the Co(OH)2 electrocatalyst with the oxygen vacancies. The synthesis method is easy and convenient, which may provide references for the other material research. I agree to publish the work after the following concerns are addressed.
1) Line 84: Double-check whether you removed Ov-Co(OH)2 or now. By my understanding, NaOH is used to remove Al rather than Ov-Co(OH)2.
2) Why do you need to do 75% iR-compensation? Any special reason for the value 75%?
3)There are two redox peaks in the CV assigned to Co(II)/Co(III) and Co(IV)/Co(III). What's the form of Co(III) and Co(IV)? Are they soluble? If they are soluble, do you think it's a concern for the durability of electrocatalyst since the applied voltage of OER is much higher than the reduction potentials of Co(II)/Co(III) and Co(IV)/Co(III).
4) SEM shows that p-Co(OH)2 has more nanoflakes. The common understanding is that the more nanoparticles you can increase the surface area. It seems that the morphologies from SEM cannot explain that the Ov-Co(OH)2 has a higher capacitive current.
Author Response
Plesae see the attacjment.

Round 2
Reviewer 1 Report
The authors did not succeed to improve the paper as much as it is needed to be published.
Reviewer 2 Report
It can be accepted for publication.
Reviewer 3 Report
I agree to publish the paper.